# Hydrological Analysis of Green Roofs Performance under a Mediterranean Climate: A Case Study in Lisbon, Portugal

Maria Luíza Santos *, Cristina Matos Silva [ID], Filipa Ferreira [ID] and José Saldanha Matos [ID]

Department of Civil Engineering and Architecture and Georesources, Instituto Superior Técnico,
University of Lisbon, CERIS, Av. Rovisco Pais, 1049-001 Lisbon, Portugal
* Correspondence: marialsantos@tecnico.ulisboa.pt

**Abstract:** In this paper, the hydrological performance of eight pilot green roofs (GR) installed in Lisbon, Portugal, under a Mediterranean climate is analyzed. The pilot units were installed at Instituto Superior Técnico campus of Lisbon University. The pilots present different plant species and different substrate types, with some of the units incorporating recycled construction and demolition waste (RCW). The green roofs pilots' hydrologic performance was evaluated through the simulation of artificial precipitation events between March 2021 and July 2021. Considering the results obtained, it can be concluded that the inclusion of RCW in the substrate composition did not hinder the development of vegetation or the hydrological performance of GR. The results showed a rainfall water retention per event ranging from 37% to 100%, with an average rainfall retention of about 81%. The runoff delay ranged from 2 to 18 min, and the peak attenuation ranged from 30 to 100%. The results indicated that previous substrate moisture strongly influences the hydrological performance of GR. As the inclusion of RCW in the substrate composition promotes a more effective drainage of the substrate during dry conditions, it is considered that RCW may have positive impacts on GR's hydrological performance.

**Keywords:** green roofs; hydrological performance; Mediterranean climate; recycled construction and demolition waste substrate; urban stormwater management

## 1. Introduction

The rapid and continuous urbanization, associated with climate change effects in terms of temperature increase, rainfall patterns and sea level rise, are causing several disturbances in urban areas, e.g., "heat island" effects, an increase in overflow discharges from combined sewer systems and more frequent flooding events.

In this context, more sustainable approaches are needed rather than simply grey techniques (e.g., larger sewers and more concrete detention tanks), including various forms of nature-based solutions, such as flooding parks, constructed wetlands, green walls, green ditches, bioswales, porous pavements and green roofs [1,2]. Those solutions fit into concepts and are known in Europe as Best Management Practices (BMPs), and they often include built components that mimic natural features, therefore constituting an environmentally beneficial approach that increases urban resilience [3,4].

The benefits from the implementation of BMPs in urban areas are diverse and include the mitigation of flood risks (due to attenuation, retardation and infiltration), climate regulation (mitigation of urban heat island effects and heatwaves) and reduction in pollution loads (associated with stormwater and overflow discharges), while promoting biodiversity and the incorporation of sociocultural services including recreational areas, as well as focusing on landscape and human well-being principles [5–7].

Green roofs (GR), which are a primary focus of this study, are to a great extent nature-based solutions [8]. GR are composed of a multilayered system that, from top to bottom, incorporates vegetation, technical substrate, filter layer, drainage layer, waterproofing layer and root barrier [9,10].

GR can be classified as intensive, semi-intensive and extensive [11]. Intensive GR have a deeper substrate and allow for a greater variety of plants (such as trees and bushes), while extensive GR are characterized by a shallow substrate and small plants, so they require less reinforced structures, compared with the ones required to support intensive GR loads [12,13]. In addition, for extensive GR, low maintenance is required and usually does not demand irrigation, making it suitable for implementation on a larger scale. Semi-intensives GR are intermediate solutions, with the substrate normally varying from 15 to 25 cm [12,14,15]. Intensive green roofs are usually designed for recreational uses and to prioritize landscaping while extensive GR are mainly developed to provide ecosystem services [16].

Substantial benefits are offered by GR, not only from an environmental perspective but also from an economic, social and aesthetic point of view [9,17], as they provide habitats for urban biodiversity [18,19] and improve city dwellers' quality of life of by creating recreational activities [20].

GR can reduce the sound exposure near or inside a building as sound transmission is mitigated by the roof system [21]: the substrate can block lower frequencies and the vegetation can block higher frequencies [19,22], even when dry [23]. Another positive impact to the buildings is the energy-related performance of GR [24], since they can absorb thermal energy and act as a thermal insulator, reducing roof surface temperature and solar heat to the covered building components, which leads to inferior cooling and heating energy requirements [19,25–27].

In addition, GR contribute to better microclimatic conditions in urban environments [28,29] as they mitigate the heat island effect by removing heat from the air through evapotranspiration and reducing the temperatures of the surrounding areas [19,30–32]. GR also improve air quality [33] as the vegetation has the ability to filter the air [10,19,34], thus decreasing traffic pollution effects [35]. Furthermore, vegetation produces oxygen and sequesters carbon dioxide, reducing greenhouse gas emissions and the urban heat island effect [36].

Another relevant benefit is the ability to regulate stormwater runoff [12,37,38] as GR retain rainwater efficiently during the early stages of rainfall events and may significantly attenuate peak runoff flows, thereby minimizing flooding risks while improving water quality [39–42].

Several papers refer to the hydrological performance of GR in different climates [43–45] or to approaches used to simulate it [46,47]. Currently, there is a consensus regarding the positive effect of green roofs in relation to rain retention; however, varying results have been reported in the literature [48]. Many factors influence the hydrological performance of a green roof, namely the composition of the substrate and plant cover, thickness of the substrate, previous moisture content, rainfall intensity and duration and GR slope [49–51]. However, the climate introduces determining factors, such as daily and seasonal temperature fluctuations, rainfall patterns, seasonal water availability, frost, wind exposure and solar radiation [52,53]. Thus, it is necessary that the design and choice of materials for the system are adequate for local conditions [53].

The reuse of materials, mainly materials obtained locally, is an opportunity to obtain more ecological green roofs [54]. Previous studies have shown that recycled materials can reduce the environmental impact associated with GR construction [54–58].

Despite the fact that many studies refer to the hydrological performance of GR, little literature exists about the hydrological performance of GR associated with the use of RCW, particularly under a Mediterranean climate. This paper focuses on the hydrological performance of GR in a Mediterranean climate, evaluating the effects of RCW use in the technical substrate.

## 2. Hydrological Performance of GR in the Mediterranean

The Mediterranean climate is characterized by wet winters with a low to moderate so-lar incidence and hot, dry summers with a high solar incidence. In these climatic conditions, rainfall is concentrated in the coldest season, often with significant rainfall intensity in short

periods of time [50,59–61]. Annual rainfall generally ranges from 300 to 900 mm/year [62]. Summer temperatures are usually hot, with an averages range from 14–25 degrees, and winter temperatures are typically mild, ranging from 7 to 13 degrees, where frost is rare [63]. The Mediterranean climate is characterized as Csa and Csb according to the Köppen climate classification [64,65].

A growing number of studies on the hydrological performance of green roofs have been developed in a Mediterranean climate [32,56,66,67]. López-Uceda et al. [67], for example, assessed the environmental risk related to the release of polluting elements in leachate from GR with RCW. Eksi et al. [66] investigated the use of RCW with a focus on vegetation development, Pushkar [56] analyzed the life cycle of GR layers and Sisco et al. [32] evaluated the efficiency of using recycled materials in GR related to plant productivity and the cooling effect of a rooftop garden. To our knowledge, there are no studies focusing on the effects of the hydrological performance of GR due to the use of RCW.

Table 1 summarizes the conditions of the experiments investigated by different authors, highlighting the climate type, the solution adopted in the experiments (e.g., the use of small-scale green roofs or real green roofs), the rainfall type (real or artificial) and the number of rainfall events.

**Table 1.** Summary of green roof hydrological performance.

| Reference | Climate Köppen Classification | Type of Green Roof | | Type of Solution | | Type of Precipitation | | Number of Events |
|---|---|---|---|---|---|---|---|---|
| | | Extensive | Intensive | Real Green Roof | Tests Beds | Real | Artificial | |
| Brandão et al. [59] | Csa | X | | | X | X | | 46 |
| Palermo et al. [50] | Csa | X | | X | | X | | 62 |
| Cristiano et al. [68] | Csa | | X | X | | X | | 6 |
| Fioretti et al. [14] | Csa | | X | X | | X | | 30 |
| Razzaghmanesh and Beecham [69] | Csa | X | X | | X | X | | 226 |
| Schultz et al. [51] | Csb | X | | X | | X | | 82 |
| BuccolaandSpolek [70] | Csb | X | | | X | | X | 2 |
| Piro et al. [71] | Csa | X | | | X | X | | 8 |
| Doménech et al. [72] | Csa | X | | X | | X | | 17 |
| Garofalo et al. [60] | Csa | X | | X | | X | | 135 |
| Palla et al. [73] | Csa | X | | X | | X | | 29 |
| Schroll et al. [74] | Csb | X | | | X | X | X | 17 |
| Soulis et al. [75] | Csa | X | | | X | X | | 45 |
| Soulis et al. [76] | Csa | X | | | X | X | | 11 |
| Rocha et al. [15] | Csa | X | | | X | | X | 3 |
| Barnhart et al. [77] | Csb | X | X | X | | X | | - |

As presented in Table 1, 75% of the studies were implemented in zones with a climate classified as Csa with the Köppen classification (hot summer Mediterranean climate), and 88% of the studies used real rainfall measurements. It is also noted that the studies using real rainfall showed a higher number of events. Most studies (88%) that evaluated the hydrological aspects of green roofs in a Mediterranean climate used extensive green roofs. In addition, all the studies based on test beds (50% of the total) presented extensive green roof features.

Table 2 shows the values of the hydrological parameters obtained by the same authors. The variable "Rainfall retention" was evaluated in 88% of the studies [19].

**Table 2.** Green roofs in Mediterranean climate: summary of reported values (hydrological parameters).

| Reference | Rainfall Retention (%) | Runoff Delay (h) | Peak Attenuation (%) | Peak Delay (h) | Runoff Coefficient | Runoff Delay (min) |
|---|---|---|---|---|---|---|
| Brandão et al. [59] | 91 | 0.45 | 99 | 0.38 | - | - |
| Palermo et al. [50] | - | - | 56 | 4.92 | 50.4% (>8 mm) | 52.1 |
| Cristiano et al. [68] | 52 and 71 | - | - | - | - | - |
| Fioretti et al. [14] | 68 | - | 89 and 74 | 2.75 | - | - |
| Razzaghmanesh and Beecham [69] | 74 and 88.6 | 3 extensive, 17 intensive | - | - | - | - |
| Schultz et al. [51] | 32.9 and 23.2 | - | - | - | - | - |
| Buccola and Spolek [70] | 20.0 to 56.0 and 36.0 to 54.0 | - | - | - | - | 3.7 to 4.9 and 5.3 to 8.1 |
| Piro et al. [71] | 57.5 | - | 72.3 | - | - | - |
| Doménech et al. [72] | 80.8 | - | - | - | 75% [1] | - |
| Garofalo et al. [60] | 80 (max) | - | - | - | 0.70 and 0.79 | - |
| Palla et al. [73] | 68 and 22 | - | 89 and 72 | 6.8 and 2.5 | - | - |
| Schroll et al. [74] | 27.2 and 64.7 | - | - | - | - | - |
| Soulis et al. [76] | 2.0 to 100.0 | - | - | - | - | - |
| Soulis et al. [76] | 42.8 (for total runoff depth) and 70.2 (peak runoff rate considered) | Aprox. 0.5 (max) | - | - | - | - |
| Rocha et al. [15] | - | - | - | - | 0.26–0.43 | - |
| Barnhart et al. [77] | 10–15 (extensive) and 20–25 (intensive) [2] | - | - | - | - | - |

[1] According to the study, if half of current conventional roofs were renovated to green roofs, runoff coefficients would be reduced to less than 75% of present ones for frequent rain events. [2] Results for the case that 10% of the watersheds analyzed were converted into green roofs.

In general, rainfall retention values were quite variable. However, in 53% of the studies registered, in at least one of the events, the rainfall retention average values were greater than 50%.

In Mediterranean climatic conditions, rainfall is concentrated in the coldest season, often with significant rainfall intensity in short periods of time [50,59,60]. It is reasonable to assume that GR can have a better hydrological performance in hot and dry summers compared to cold and wet winters, since this performance is directly related to the previous degree of humidity of the substrate. In the rainy seasons (winter in the Mediterranean), the substrates will be frequently wet and close to saturation, and therefore will be able to retain less rainwater, resulting in a humbler hydrological performance [78].

## 3. Materials and Methods

### 3.1. Site Description

The construction of the first four GR test beds occurred in December 2020, and the other four test beds with RCW occurred in May 2021 in an area located in the civil engineering building (38.736896, −9.140335) at the Instituto Superior Técnico campus (IST), as presented in Figure 1. The fieldwork with artificial rainfall on the GR test beds was conducted from March to July 2021.

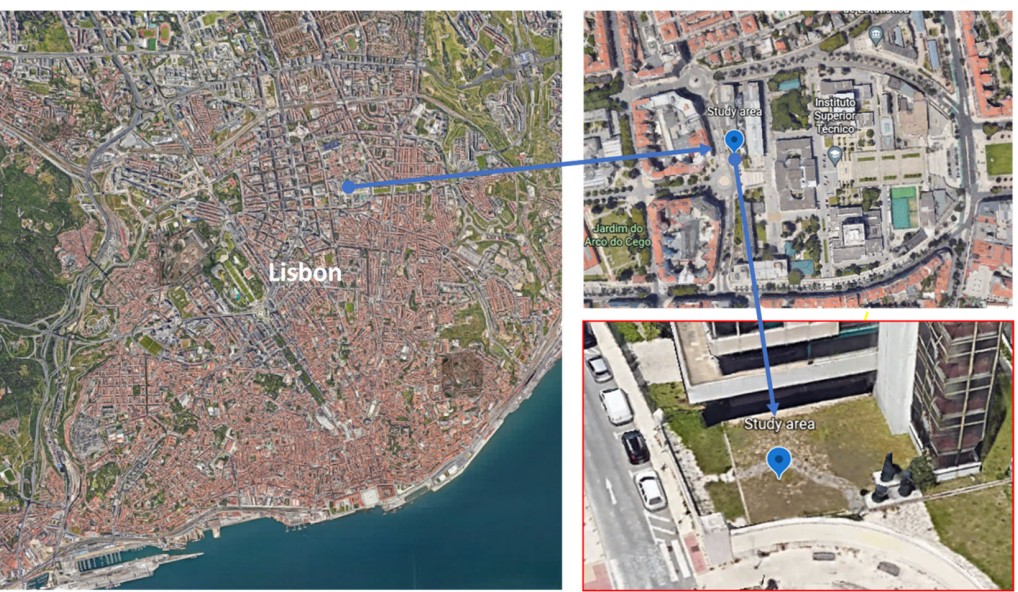

**Figure 1.** Location of the green roof test beds (pilot units) at Instituto Superior Técnico campus, Lisbon, Portugal.

### 3.2. Test Beds Characteristics

For this study, eight extensive green test beds (named from G1 to G8) were built using 676 L HDPE pallet boxes, with external dimensions of 1.20 × 1.00 × 0.78 m (internal dimensions are 1.12 × 0.92 × 0.61 m). The test beds were elevated (15 cm from the ground), supported by cement blocks, as illustrated in Figure 2.

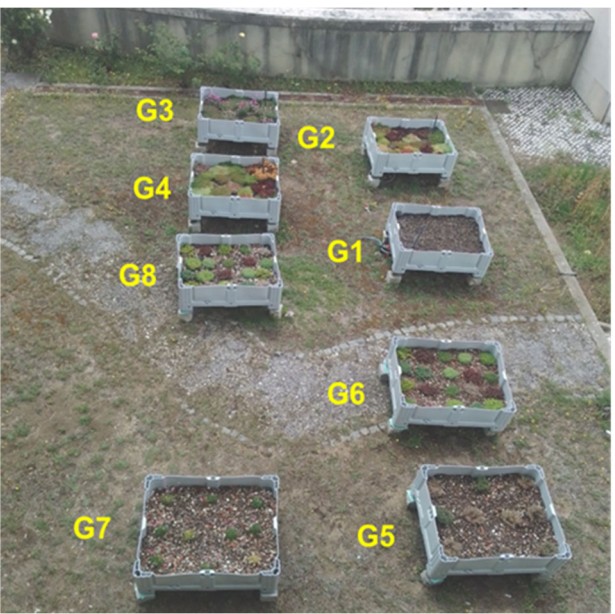

**Figure 2.** View of the eight green roof tests beds.

The GR configuration was extensive with a multilayered structure, as presented in Figure 3. The material of the boxes was waterproof. A bottom layer with a retention and protection blanket for the pallet box was installed, consisting of a geotextile, SSM45 ZinCo, functioning as a mechanical barrier to the action of the roots. The next layer consisted of the drainage element, composed of modular drainage panels from the brand Floradrain FD25, ZinCo International. These panels, made of recycled polyolefin material, with a

compressive strength of 270 kN/m$^2$ and a thickness of 25 mm, are recommended for roofs of extensive types [79].

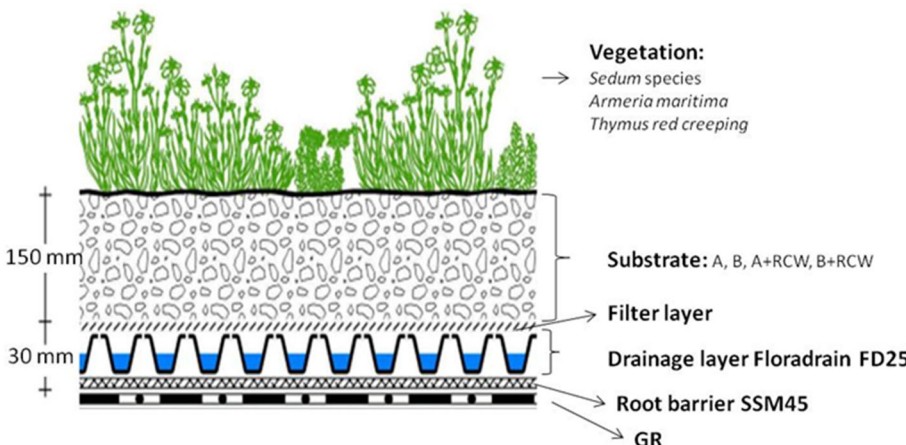

**Figure 3.** Constituent layers of test beds.

At the top of the drainage layer, the filtering layer was installed consisting of the filter system from the ZinCo brand, with the aim of preventing fine particles and sediments from the substrate layer from passing to the lower layers of the draining system.

The test beds were built as described in Mendonça [80], with varying substrate types, vegetation types and presence or absence of recycled construction and demolition waste materials (RCW) mixed with the substrate. Figure 4 shows the technical substrate and the RCW, which consisted of a mixture of different proportions of concrete, bricks, tiles and ceramics before being homogenized. The adopted combinations are described in Table 3. Two different substrate types were chosen (substrate A with 21% of organic matter content and substrate B with 41% of organic matter content), as well as 10 different vegetation species. Whenever RCW was included, the proportion of technical substrate was 66.6% of the total.

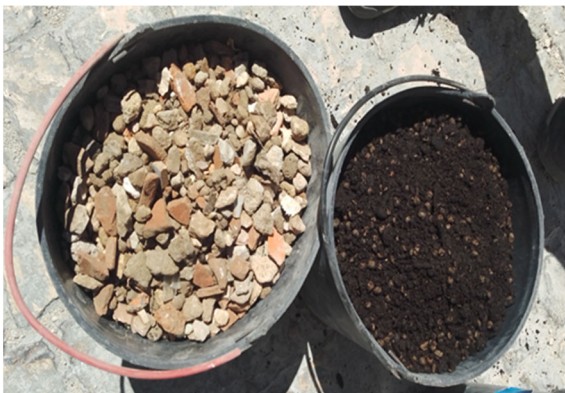

**Figure 4.** Technical substrate and RCW used before mixing.

Plants species adapted to a Mediterranean climate were selected. The species used were *Sedum album*, *Sedum sexangular*, *Sedum spurium*, *Sedum spurium tricolor*, *Sedum coral reef*, *Sedum oreganum*, *Sedum forsteriamum*, *Armeria Maritima* and *Thymus red creeping e Rosmarinus officinalis*.

**Table 3.** Characteristics of the green roof test beds.

| Test Bed | Substrate | | | | Vegetation | | | | | | | | | |
|---|---|---|---|---|---|---|---|---|---|---|---|---|---|---|
| | A | B | A + RCW | B + RCW | *Sedum Album* | *Sedum Coral Reef* | *Sedum Forsterianum* | *Sedum Oreganum* | *Sedum Sexangular* | *Sedum Spurium* | *Sedum Spurium Tricolor* | *Arneria Maritima* | *Thymus Red Creeping* | *Rosmarinus Officinalis* |
| G1 | X | | | | | | | | | | | | | X |
| G2 | X | | | | | | X | X | X | X | X | | | |
| G3 | X | | | | | | | | | | | X | X | |
| G4 | | X | | | | | X | X | X | X | X | X | | |
| G5 | | | X | | | | | | | | | | | X |
| G6 | | | X | | | | X | X | X | X | X | | | |
| G7 | | | X | | | | | | | | | X | X | |
| G8 | | | | X | X | X | | X | X | X | X | | | |

### 3.3. Data Collection and Analysis

Tables 4 and 5 detail the substrate characteristics in different test beds, obtained by laboratory analysis. Regarding the granulometry of the substrates, it can be concluded that the substrates with RCW had a higher percentage of fine fraction and a lower percentage of coarse fraction than the substrates without RCW. The pH of all substrates was close to neutrality, but B and B + RCW were more basic. The concentration of nitric nitrogen (the form most used by plants) was substantially higher in A + RCW, while extractable phosphorus was higher in B. Extractable potassium was higher in substrates with RCW. The percentage of organic matter was higher in B and, consequently, B + RCW also presented a higher percentage than A + RCW. The percentage of dry matter was higher in A + RCW and was relatively similar in the other three substrates. The highest densities were shown by the substrates containing RCW due to the addition of the recycled materials. The hydraulic conductivity of the saturated soil was higher in substrate B and quite similar in the substrates with RCW.

**Table 4.** Granulometric characteristics of substrates of the green roof test beds.

| | Granulometry | Subst. A | Subst. B | Subst. A + RCW | Subst. B + RCW |
|---|---|---|---|---|---|
| | Fine fraction (%) | 26.2 | 17.8 | 37.2 | 34.5 |
| | Coarse fraction (%) | 73.8 | 82.2 | 62.8 | 65.5 |
| Granulometry (<2 mm) | Coarse sand (g kg$^{-1}$) | 278.2 | 208.0 | 208.0 | 304.7 |
| | Fine sand (g kg$^{-1}$) | 172.0 | 117.3 | 117.3 | 162.1 |
| | Slime (g kg$^{-1}$) | 343.5 | 476.1 | 476.1 | 333.6 |
| | Clay (g kg$^{-1}$) | 206.3 | 198.7 | 198.7 | 199.6 |

The substrate moisture was monitored before each event with the HS2 HYDROSENSE II sensor. Although this sensor was designed for use in agricultural soils, the measurement technique underlying the instrument supports other potential applications [81]. Nevertheless, as the soil is not homogeneous, in each test bed the humidity was measured at five points and then the average value for each GR pilot unit test was determined.

The climatic data were registered by a meteorological station located at Instituto Superior Técnico (more information is available at meteo.tecnico.ulisboa.pt).

**Table 5.** Substrates characteristics of the green roof test beds.

| Parameters | Subst. A | Subst. B | Subst. A + RCW | Subst. B + RCW |
|---|---|---|---|---|
| pH ($H_2O$) | 7.1 | 8.0 | 7.0 | 7.7 |
| Nitric nitrogen (mg/L) | 3.3 | 5.6 | 431 | 7.7 |
| Extractable phosphorus (mg/L) | <1.0 | 9.4 | <1.0 | 2.4 |
| Extractable potassium (mg/L) | 35.9 | 91.3 | 207 | 173 |
| Organic matter (%) | 20.9 | 41.4 | 14.9 | 34.6 |
| Dry matter (%) | 54.4 | 59.0 | 71.8 | 55.5 |
| Density (g/cm$^3$) | 0.53 | 0.47 | 0.78 | 0.71 |
| Hydraulic conductivity of saturated soil (cm/h) | 124.1 | 361.6 | 270.3 | 249.8 |

In this experimental study, nine artificial rainfall events were produced using a garden shower. Through manual control of the water flow, different rainfall intensity events were simulated. For simulation of the most intensive rainfall events, two showers were used on each pilot unit. Due to equipment limitations, Weak rainfall events were not simulated (rainfall intensity less than 2 mm/h).

The time interval from the beginning of the rainfall simulation to the beginning of runoff was recorded, and the total runoff was measured with 5 min intervals.

As for the maintenance of the test beds, weeds were removed weekly, and during dry periods, the beds were watered once a week.

The vegetation cover was monitored through weekly photographic records taken from January 2021 to January 2022. The photos were processed using the image analysis software ImageJ, as described by Liberalesso et al. [13], Perez et al. [16] and Sněhota et al. [82].

*3.4. Data Analysis*

To evaluate the hydrological performance of the installed GR, the following variables were considered, as they are representative of the hydrological cycle and commonly studied [19]:

- Rainfall water retention (R, %), which is the difference between the total incoming rainfall depth and the total runoff depth divided by the total rainfall depth.
- Runoff delay (RD, minutes), which is the time difference between the beginning of rainfall and the beginning of runoff.
- Peak attenuation (PA, %), which is the difference between rainfall and runoff peaks (maximum values registered for 10 min durations) divided by the rainfall peak.

**4. Results and Discussion**

*4.1. Vegetation Development*

Figure 5 shows a graph with the evolution of the vegetation cover in the GR beds G1 to G8, expressed in % of covered area, while Figure 6 presents images of the evolution of the vegetation cover.

The *sedum* species adapted better to the Mediterranean conditions and presented greater growth.

From what was observed, the vegetation planted in substrates with RCW did not present additional development difficulties when compared to vegetation planted in the substrates without RCW. On the contrary, in test bed G6 (A + RCW substrate and *sedum* vegetation), plants presented a rapid development: a month after plantation, G6 showed a vegetation cover of more than 90%. A similar result was found in Mickovski et al. [83], who also developed a study with small-scale pilot GR, in which they investigated the potential for using recycled construction waste in the substrates. Mickovski et al. [83] reported that *sedum* and grass species could effectively establish rather well in a substrate mixture with construction waste.

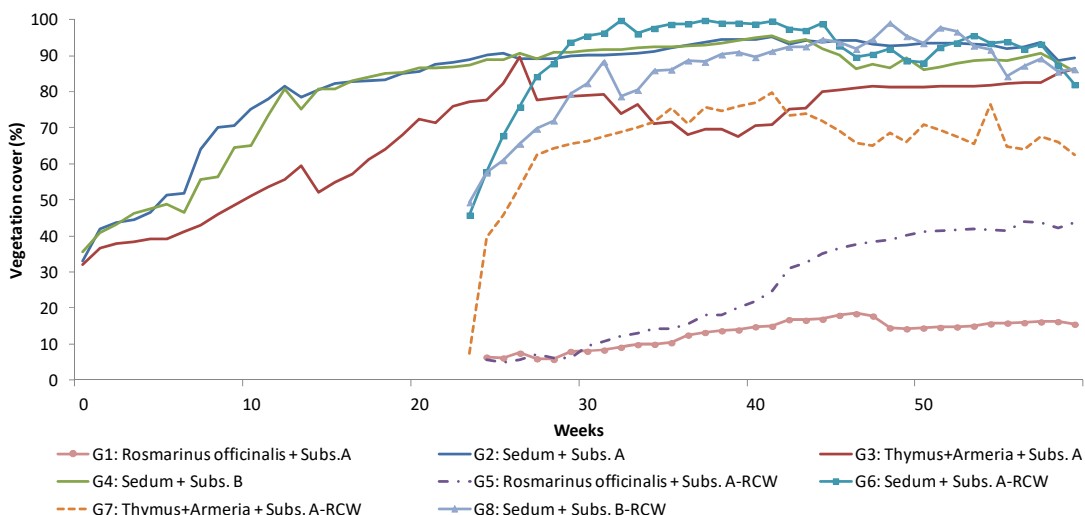

**Figure 5.** Vegetation cover expressed as a % of G1 to G8 beds (from the first week to week 50).

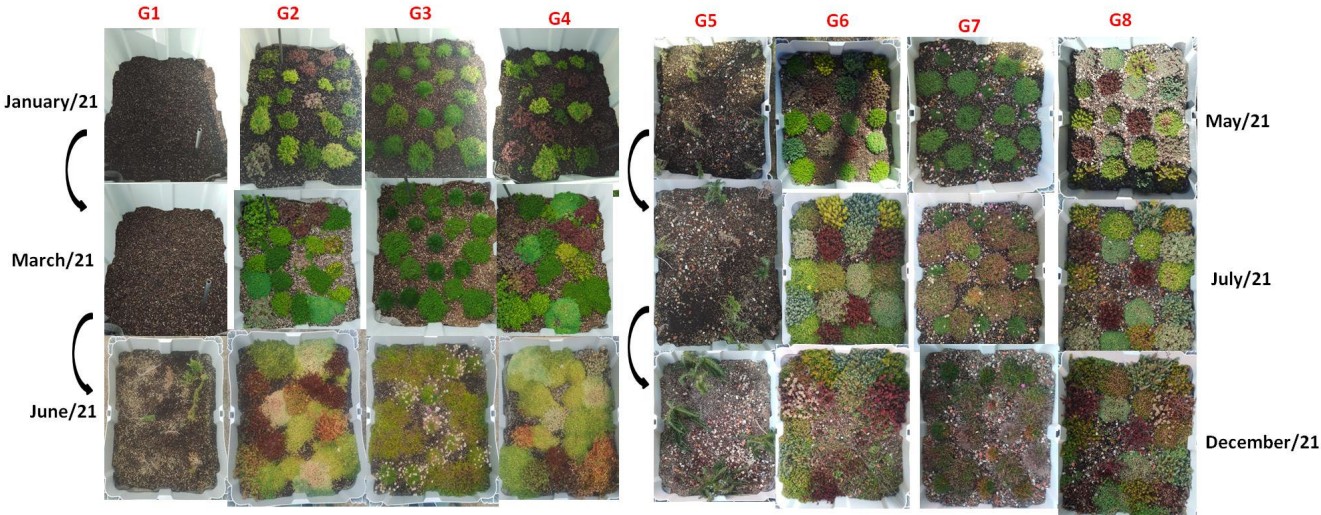

**Figure 6.** Evolution of vegetation growth of the test beds between January 2021 and December 2021.

By late summer 2021, all the GR containing *sedum* species presented vegetation covers above 90%. A similar result was found in the study conducted by Gong et al. [84], where values over 95% vegetation cover in the pilot extensive GR with different species of sedum were reported in a typical continental monsoon climate.

Di Miceli et al.'s [85] results revealed that *sedum* species are suited to use in green roofs in a Mediterranean climate, managing to grow with a low input maintenance. The same was verified in this investigation, showing a greater adaptability of the *sedum* species in relation to other species.

Brandão et al. [59], at the end of their experiment carried out in Lisbon, reached a plant cover with *Rosmarinus officinalis* of 28.9%, which is not very different from the plant cover reached with the same species in this investigation. G1 obtained a vegetation cover of approximately 15.6%, while G5 presented a vegetation cover of approximately 43.8%. In this case, in the test bed with substrates containing RCW, the plants showed a slightly better development. In fact, *Rosmarinus officinalis* seedlings were planted in G1 and G5 on the same day and, at the end of the evaluated period, G5 presented a significantly higher plant cover.

It should be mentioned that a punctual vegetation cover decrease was observed in the test beds, either due to the intentional removal of dry twigs and leaves or due to



birds' interferences. In addition, as expected, in the winter period, a lower growth or even decrease in some plants was observed.

Differences regarding substrate A and B did not seem to influence plant development. G2 and G4 had the same vegetation composition (*sedum* species) but with different substrates (G2—substrate A and G4—substrate B), and at the end of the studied period, they presented a very similar vegetation cover (both above 90% in January 2022). According to the results, neither the vegetation development nor the type of species seemed to influence the hydrological efficiency of GR, as also reported by Silva et al. [19], who suggested that the type and origin of the vegetation might be more relevant that the plant coverage (%).

### 4.2. Artificial Rainfall

Table 6 shows the characteristics of the simulated artificial rainfalls: precipitation intensity, event duration and frequency, rainfall depth and rainfall classification as defined by the Portuguese Institute of Sea and Atmosphere (IPMA). According to the IPMA, rainfall intensity may be classified into four categories: Weak (<2 mm/h); Moderate (between 2–10 mm/h); Strong (between 10–50 mm/h); and Violent (>50 mm/h). Rainfall intensity was determined using Lisbon rainfall intensity–duration–frequency (IDF) curves [86,87].

**Table 6.** Test beds—artificial rainfall characterization.

| Events | Test Beds | Date | Flow Rate (L/s) | Rainfall Intensity (mm/h) | Rainfall Duration (minutes) | Rainfall Frequency | Rainfall Depth (mm) | Rainfall Classification |
|---|---|---|---|---|---|---|---|---|
| 1 | G1–G4 | 06/04/2021 | 0.0480 | 168.20 | 10 | >50 years | 28.03 | Violent |
| 2 | G1–G4 | 13/04/2021 | 0.0120 | 42.45 | 15 | 2 years | 10.61 | Strong |
| 3 | G1–G4 | 20/04/2021 | 0.0035 | 12.40 | 15 | 3 times per year | 3.10 | Strong |
| 4 | G1–G4 | 27/04/2021 | 0.0024 | 8.39 | 20 | 7 times per year | 2.80 | Moderate |
| 5 | G1–G4 | 11/05/2021 | 0.0066 | 23.16 | 20 | 1 years | 7.72 | Strong |
| 6 | G1–G8 | 25/05/2021 | 0.0101 | 35.50 | 20 | 2 years | 11.82 | Strong |
| 7 | G1–G8 | 01/06/2021 | 0.0128 | 44.96 | 20 | 5 years | 14.99 | Strong |
| 8 | G5–G8 | 08/06/2021 | 0.0182 | 63.59 | 20 | 20 years | 21.20 | Violent |
| 9 | G5–G8 | 09/06/2021 | 0.0182 | 63.59 | 20 | 20 years | 21.20 | Violent |

The first event simulated the most intensive rainfall (Violent classification) with a duration of 10 min. Events 2 and 3 (with Strong rainfalls) lasted for 15 min, as simulated in the study by Buccola and Spolek [70]. From the fourth event on, durations of 20 min were considered, which was the time used in the investigation by Kemp, Hadley and Blanuša [88]. The lowest rainfall intensity was attained in event 4 (8.39 mm/h), corresponding to a Moderate rainfall. Palermo et al. [50], in a study using a Mediterranean climate with real precipitation events, just selected for further analysis rainfall events with a depth of more than 2 mm. In the present study, all simulated events presented a rainfall depth higher than 2 mm. The maximum rainfall recorded in the study by Palermo et al. [50] was 167.60 mm/h, comparable to the rainfall intensity of event 1 (168.20 mm/h).

### 4.3. Hydrological Performance

In GR, rainfall retention is namely influenced by (i) the rainfall intensity and duration [89–91]; (ii) the substrate depth and other physical properties [51,89,90,92]; (iii) the vegetation [93]; and (iv) the humidity of the substrate, which can be related to the saturation point [19,51,89,94]. In addition, the interactions between vegetation and substrate influence water retention, since root systems change the physical conditions of the substrate, namely the soil density and global porosity [15].

Table 7 presents the results of the average moisture of the test beds substrates. Given the fact that the climatic conditions before each rain event were similar in all the test beds, the differences in the substrate moisture were likely due to the substrate composition or vegetation. G4 and G2 had the same type of vegetation but different substrates. In most

events, G4 showed lower substrate moisture than G2 (except in event 6, where the substrate moistures were similar). This may indicate that substrate B, which has a higher fraction of organic matter, has a greater capacity to retain water, as referred to by Spolek et al. [95].

**Table 7.** Substrate moisture of the test beds before the simulated rainfall event.

| Events | Substrate Moisture before the Rainfall Event (%) | | | | | | | |
|---|---|---|---|---|---|---|---|---|
| | **G1** | **G2** | **G3** | **G4** | **G5** | **G6** | **G7** | **G8** |
| 1 | 41.7 | 23.2 | 18.4 | 11.5 | - | - | - | - |
| 2 | 39.4 | 33.0 | 31.5 | 19.4 | - | - | - | - |
| 3 | 24.2 | 23.8 | 19.7 | 15.6 | - | - | - | - |
| 4 | 26.9 | 27.9 | 25.4 | 21.6 | - | - | - | - |
| 5 | 19.1 | 30.6 | 38.1 | 29.8 | - | - | - | - |
| 6 | 15.3 | 14.2 | 22.8 | 14.4 | 13.5 | 6.3 | 19.3 | 9.6 |
| 7 | 12.7 | 18.2 | 26.2 | 14.2 | 14.5 | 6.4 | 13.2 | 9.9 |
| 8 | - | - | - | - | 7.5 | 5.3 | 7.3 | 7.7 |
| 9 | - | - | - | - | 26.3 | 29.6 | 31.3 | 22.0 |

For events 6 to 9, test beds G5–G8 (including RCW) presented lower substrate moistures than G1–G4. It can be concluded that the inclusion of RCW in the substrate composition promoted faster drainage of the substrate, which favored GR's hydrological performance since, as verified in the literature [78,89,96], lower humidity values in the substrate are associated with greater rainfall retention efficiencies.

Table 8 shows the rainfall retention in each precipitation event. Figure 7 shows the relationship between the initial moisture content, rainfall intensity and water retention for each simulation.

**Table 8.** Rainfall average retention in each test bed and average retention per event.

| Events | Percentage Absorbed by the Substrate (%) | | | | | | | | Average Event Retention |
|---|---|---|---|---|---|---|---|---|---|
| | **G1** | **G2** | **G3** | **G4** | **G5** | **G6** | **G7** | **G8** | **(%)** |
| 1 | 55.4 | 72.4 | 84.5 | 68.9 | - | - | - | - | 70.3 |
| 2 | 76.2 | 45.1 | 62.2 | 55.9 | - | - | - | - | 59.9 |
| 3 | 100.0 | 100.0 | 100.0 | 100.0 | - | - | - | - | 100.0 |
| 4 | 100.0 | 100.0 | 100.0 | 100.0 | - | - | - | - | 100.0 |
| 5 | 100.0 | 23.4 | 64.0 | 81.4 | - | - | - | - | 67.2 |
| 6 | 100.0 | 100.0 | 100.0 | 100.0 | 100.0 | 100.0 | 100.0 | 100.0 | 100.0 |
| 7 | 100.0 | 100.0 | 100.0 | 100.0 | 100.0 | 100.0 | 100.0 | 100.0 | 100.0 |
| 8 | - | - | - | - | 98.1 | 98.7 | 100.0 | 98.4 | 98.8 |
| 9 | - | - | - | - | 38.4 | 35.5 | 41.6 | 32.3 | 37.0 |

According to the data shown in Table 8, in some events (namely events 3, 4, 6 and 7), the pilot GR were able to retain all the rainfall, and no runoff occurred. In event 8, which was only simulated in the test beds with RCW (G5 to G8), the test beds presented high runoff retentions, all above 98%. However, in the next day's event (event 9), with the same rainfall intensity and event duration, the retentions were much lower, as the substrate moisture was already high due to the previous event. Consequently, the substrate reached saturation faster and thus retained less water. This effect can also be seen in Figure 6.

Villarreal and Bengtsson [91] reported that higher runoff retentions were obtained in low-intensity rainfall events. In the present study, this was also verified in the case of events 3 and 4, with the lowest intensity where total retention was achieved. The influence of rainfall intensity on the performance of GR was also highlighted as important and significant in studies by [89,95–97]. In event 8, a high retention was also obtained (average of 98.8%) with a much higher intensity (63.59 mm/h). In this case, the high retention was

probably due to the low initial substrate moisture of the substrates, as shown in Table 7. A similar result on rainfall retention related to previous substrate moisture was found by [89].

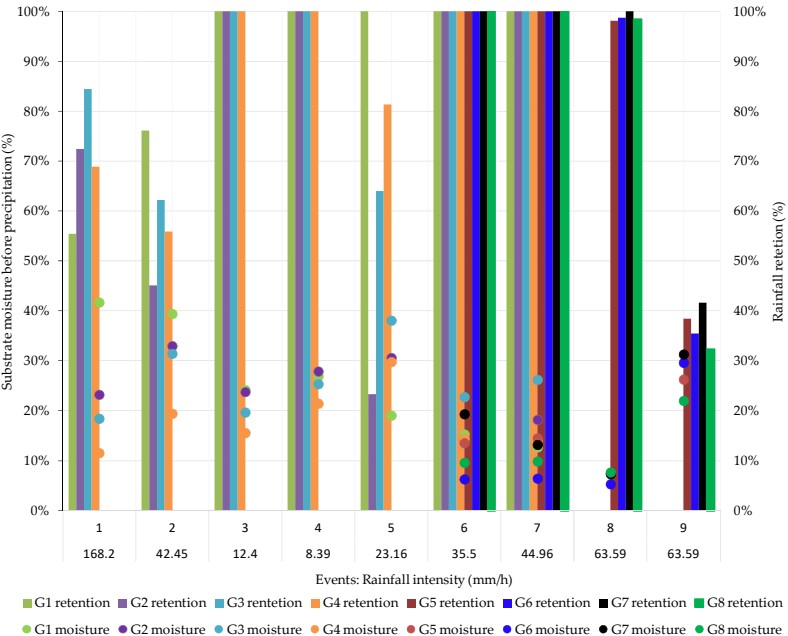

**Figure 7.** Relationship between initial moisture content, rainfall intensity and rainfall retention.

Approximately 44% of the rainfall simulation events did not generate runoff. This observation is similar to the one reported by Brandão et al. [59], who observed that more than 1/3 of rainfall events did not produce runoff. In addition, the fact that total retention was observed in the more frequent events demonstrates the advantages of adopting GR solutions in a Mediterranean climate.

The average retention of all GR in the nine events was about 81%. In the literature review for the Mediterranean climate, different average rainfall retentions were reported, as can be seen in Table 2. In the present investigation, the substrate composition did not seem to have a large impact on rainfall retention: in GR without RCW, the average rainfall retention was about 85%, while in GR with RCW, it was about 84%. Furthermore, the presence of RCW did not affect the development of the plants in test beds G5 to G8.

Another important aspect about rainfall water retention is the fact that, although runoff was measured for a period five times longer than the duration of the event (90 min versus 10 to 20 min), the total accumulated effluent volume of the beds was significantly lower than the accumulated inflow volume. This can be illustrated by Figure 8, which presents an accumulated rainfall and runoff volume for different test beds during event 2.

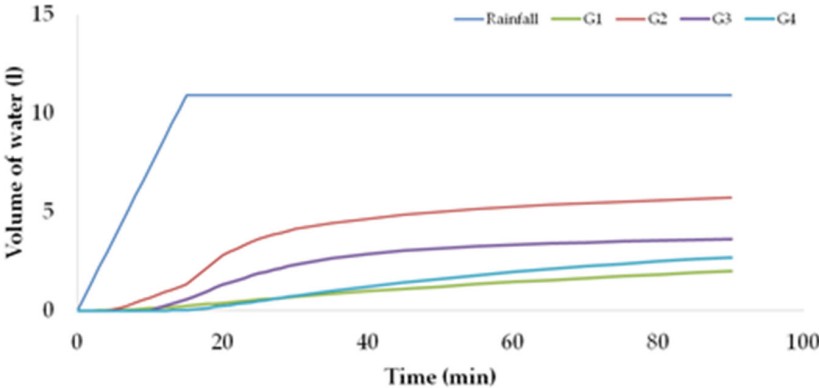

**Figure 8.** Accumulated rainfall and runoff volumes for different test beds during event 2.

The runoff delay, observed in events 1, 2, 5, 8 and 9, which produced runoff, is shown in Table 9.

**Table 9.** Runoff delay of each event.

| Events | Runoff Delay (hour) | | | | | | | |
|---|---|---|---|---|---|---|---|---|
| | G1 | G2 | G3 | G4 | G5 | G6 | G7 | G8 |
| 1 | 2.4 | 8.8 | 10.7 | 7.4 | - | - | - | - |
| 2 | 2.2 | 4.1 | 14.5 | 13.0 | - | - | - | - |
| 5 | No runoff occurred | 9.3 | 17.7 | 4.9 | - | - | - | - |
| 8 | - | - | - | - | 14.7 | 20.7 | No runoff occurred | 15.7 |
| 9 | - | - | - | - | 5.6 | 6.4 | 5.0 | 6.6 |

In events 1 and 2, the runoff delay was significantly shorter in G1 than in the other test beds, as the initial substrate moisture of G1 was higher. In event 5, G1 presented the lowest initial substrate moisture of the four test beds, and it was the only one that retained all the simulated rainfall (no runoff occurred). In the last two events, 8 and 9, despite having the same rainfall intensity, very different runoff delays were observed. This can be explained by the fact that, in the first event, the substrate was drier.

The results for the first events, 1 and 2, suggest that G3 may present a greater capacity to retain water, since it showed the longest time before runoff started. As G3 has the same substrate as G1 and G2 but a different vegetation (*Thymus red creeping* and *Armeria maritima*), the results may suggest some impact of the vegetation on this time lag. Nevertheless, it is considered that more studies are needed to confirm this observation.

The experimental studies evaluating runoff delay showed rather different results (Table 2). The present study results are closer to the ones presented by Buccola and Spolek [70].

The peak attenuations estimated for each event and test bed are presented in Table 10. Events 3, 4, 6 and 7 retained the total rainfall and therefore were not included in this table. To illustrate the variation in runoff with time, Figure 9 (for event 2) is presented.

**Table 10.** Peak attenuation (%) for each event and test bed.

| Events | Peak Attenuation (%) | | | | | | | |
|---|---|---|---|---|---|---|---|---|
| | G1 | G2 | G3 | G4 | G5 | G6 | G7 | G8 |
| 1 | 93 | 76 | 93 | 84 | - | - | - | - |
| 2 | 94 | 59 | 79 | 93 | - | - | - | - |
| 5 | 100 | 30 | 63 | 91 | - | - | - | - |
| 8 | - | - | - | - | 98 | 99 | 100 | 98 |
| 9 | - | - | - | - | 76 | 55 | 70 | 58 |

The results demonstrated that the peak flow attenuation of extensive GR in a Mediterranean climate is significant, as was also concluded by Piro et al. [71]. Considering only the events in which runoff occurs, 60% of the events had an average peak attenuation above 80%. The results are not very different from the ones presented by Brandão et al. [59], with 75% of the events presenting peak flow attenuations higher than 89%.

In the first events (1, 2 and 5), G2 had the lowest peak attenuation. G2 had the same vegetation as G4 (*sedum*) but different substrates. In agreement with the analysis previously mentioned, substrate B seemed to have a greater capacity to retain water than substrate A, which also resulted in greater peak flow attenuations.

The influence of previous substrate moisture on the peak attenuation, as also reported by Palermo et al. [50], was confirmed in the present study.

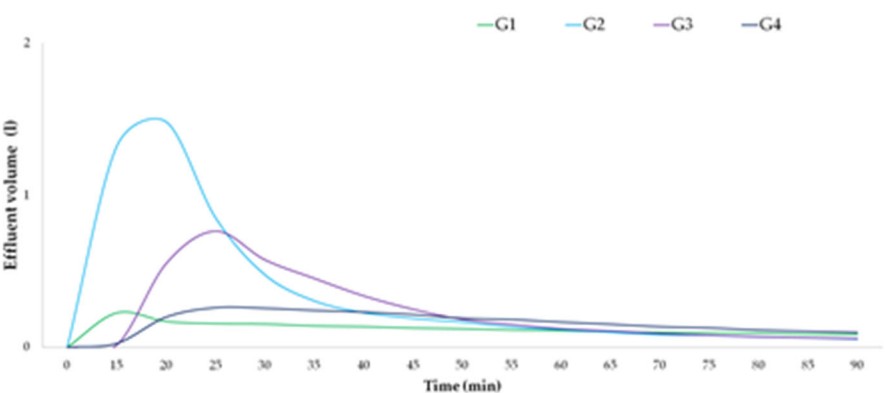

**Figure 9.** Event 2 (42.45 mm/h) effluent flow variation.

## 5. Conclusions

Green roofs are source control techniques that have been implemented in urban areas due to their multiple benefits, such as the increase in the aesthetic and economic value of buildings and the mitigation of heat island effects and biodiversity losses. Green roofs are emerging as alternatives for better stormwater management as they increase pervious areas, local water storage and evapotranspiration, contribute to peak flow attenuation and minimize overflows and flooding events.

This study evaluates the hydrological performance of eight extensive GR installed under a Mediterranean climate. The test beds included different substrate compositions, some incorporating RCW, and various vegetation species, and nine rainfall simulations were performed.

Considering the results obtained, it can be concluded that the inclusion of RCW in the substrate composition did not hinder the development of vegetation or the hydrological performance of GR. On the contrary, in some test beds containing RCW, plants showed a better development. *Sedum* and *Romarinnus officinalis* species in particular are considered suited to use in GR with RCW under a Mediterranean climate, managing to grow with low input maintenance.

Furthermore, a technical substrate mixture comprising RCW (in the ratio of 33% RCW to 66% technical substrate) can provide significant water retention, runoff delay and peak flow attenuation. The results showed a rainfall retention per event ranging from 37% to 100%, with an average rainfall retention of about 81%. The presence of RCW did not seem to have an impact on the GR rainfall retention, indicating that recycled construction materials can be suited for GR substrate composition.

The peak flow attenuation ranged from 30 to 100%. The results indicated that previous substrate moisture strongly influenced the hydrological performance of GR. As the inclusion of RCW in the substrate composition promoted a more effective drainage of the substrate, it is considered that RCW has a positive impact on GR's hydrological performance.

Nevertheless, as only artificial rainfalls were evaluated and the GR were still relatively young, and vegetated GR substrate undergoes various physical–chemical changes over time [94], it is considered that additional studies should take place to better quantify the impacts of RCW and vegetation on the hydrological performance of GR.

**Author Contributions:** Conceptualization, M.L.S., F.F., C.M.S. and J.S.M.; methodology, M.L.S. and F.F.; supervision, C.M.S., F.F. and J.S.M.; visualization and writing—original draft preparation, M.L.S.; writing—review and editing, C.M.S., F.F. and J.S.M. All authors have read and agreed to the published version of the manuscript.

**Funding:** The authors are grateful for FCT's, Fundação para a Ciência e Tecnologia, support through funding UIDB/04625/2020 from the research unit CERIS. This work was supported by the FCT through the research project GENESIS (PTDC/GESURB/29444/2017) and the research unit UID/AGR/04129/LEAF.

**Institutional Review Board Statement:** Not applicable.

**Informed Consent Statement:** Not applicable.

**Data Availability Statement:** The results data presented in this work are available on request from the corresponding author if no sensible data are involved regarding the real case study.

**Acknowledgments:** The authors acknowledge the support from CERIS and Instituto Superior Técnico. We also thank FCT for the scholarship (PD/BD/150559/2019) granted to Maria Luíza da Cunha Oliveira Santos.

**Conflicts of Interest:** The authors declare no conflict of interest.

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
