# Peer review of "Hydrological Analysis of Green Roofs Performance under a Mediterranean Climate: A Case Study in Lisbon, Portugal"

_sustainability, doi:10.3390/su15021064_

Round 1
Reviewer 1 Report
Authors have carried out to study on “Hydrological analysis of green roofs performance in a Mediterranean climate: a case study in Lisbon, Portugal”. The study contains eight pilot green roofs with different subgrades.
The following are the Observations need to be addressed:
· Typographical errors are mentioned in the Manuscript need to be corrected.
· Add the limitations of the study also.
· Kindly add the section on instruments used and their measurement ability.
· The results and discussion should focus on the effect of grass type, subgrade type and the rainfall intensity effect on the hydrological properties.
· The conclusions are too general and contain discussion part, specific outcome of the study need to be mentioned.
· Few places references are missing.

Author Response
Response to Reviewer 1 Comments
The authors wish to express gratitude to the reviewers for their time and effort in reviewing our manuscript.
As suggested by Reviewer 2, the paper was completely restructured and rewritten with a new perspective. Many changes were implemented, some to correct minor typographical errors, but most regarding the restructure of the manuscript. Chapter 1 (Introduction) was improved to include a general introduction and more background information on GR; to refer to other research papers and provide more information on GR specifications, variations, and benefits; and to introduce the important aspects of our study. A second chapter was created (2. Hydrological performance of GRs in the Mediterranean) referring to the Mediterranean climate and summarizing the existing studies on the hydrological performance of GRs in the Mediterranean.
Chapter 3 (Materials and Methods) was reorganized in three subchapters referring to site description, test beds characteristics, data collection and analysis and data analysis. This chapter presents the study site and experimental set up, the substrate characterization and composition, the methods and instruments used in the study, including their main limitations, and the variables considered to evaluate the hydrological performance of the installed GR.
Chapter 4 was rewritten to present the “Results and Discussion”, including the appropriate referencing and discussion points focusing on the effect of vegetation, substrate composition, previous moisture content and rainfall characteristics on the hydrological performance of GR.
Finally, in Chapter 5 (Conclusions) a summary of the study and it’s relevance is presented, evidencing the specific outcome of the investigation.
It should be noted that a thorough bibliographic review was done resulting in the inclusion of more than 50 new references in the paper.
As required, we have submitted a document entitled “compare doc.docx” that compares the initial and the final versions of the paper, with track changes turned on. In addition, due to the wide range of changes in the document, and to simplify the Reviewers understanding of the implemented modifications, we decided to prepare a file (“_sustainability-New V10_red changes.docx”) that consists of a final version of the paper, where the most important additions are highlighted in red text.
Point 1: Typographical errors are mentioned in the Manuscript need to be corrected.
Thanks for the suggestions. Mentioned typographical errors have been corrected.
Point 2: Add the limitations of the study also.
Thanks for your comment. We have added the following paragraph in the conclusion that addresses the limitations of the study.
“Nevertheless, as only artificial precipitations were evaluated and the GR were still relatively young (and vegetated GR substrate undergoes various physical-chemical changes over time [94], additional studies should take place to quantify the impacts of RCW and vegetation in the hydrological performance of GR.”
Point 3: Kindly add the section on instruments used and their measurement ability
Thanks for your suggestion. Information about the instruments used and their measurement ability was included in the section "3.3. Data collection and analysis", as exemplified in the paragraphs below:
“The substrate moisture was monitored before each event with the HS2 HYDROSENSE II sensor, by inserting the sensor vertically into the substrates of all test beds. Although this sensor was designed for use in agricultural soils, the measurement technique underlying the instrument supports other potential applications [81]. Nevertheless, as soil is not homogeneous which may lead to difficulties in accurate measurements, in each test bed, the humidity was measured at five points and then the average value for each GR pilot unit test was determined.
The climatic data was registered by a meteorological station located at Instituto Superior Técnico (more information is available in meteo.tecnico.ulisboa.pt).
In this experimental study, nine artificial rainfall events were produced using a garden shower. Through manual control of the water flow, different rainfall intensities events were simulated. For simulation of the most intensive rainfall events, two showers were used on each pilot unit. However, due to limitations related to this equipment, it was not possible to simulate weak rainfall events (rainfall intensity less than 2 mm/h).”
Point 4: The results and discussion should focus on the effect of grass type, subgrade type and the rainfall intensity effect on the hydrological properties.
Thank you for your professional comment. We improved the discussion of the results (in Chapter 4) considering your suggestion and more references have been added to support the presented discussion.
Point 5: The conclusions are too general and contain discussion part, specific outcome of the study need to be mentioned.
Thanks for your pertinent comment. The conclusion has been rewritten, presenting the specific outcomes of this study. The following paragraphs emphasise the main conclusions:
“Considering the results obtained, it can be concluded that the inclusion of RCW in the substrate composition did not hinder the development of vegetation or the hydrological performance of GR. On the contrary, in some test beds containing RCW, plants have shown a better development. Sedum and Romarinnus officinalis species in particular are considered suited to use in GR with RCW under Mediterranean climate, managing to grow with low input maintenance.
Furthermore, technical substrate mixture comprising RCW (in the ratio of 33% RCW to 66% technical substrate) can provide significant rainfall water retention, runoff delay and peak attenuation. Results showed rainfall retention per event ranging from 37% to 100%, with an average rainfall retention of all events of about 81%. The presence of RCW did not have an impact on the GR rainfall retention, indicating that recycled construction materials can be reused in GR substrate composition.
The runoff delay ranged from 2 to 18 minutes and peak attenuation ranged from 30 to 100%. Results indicate that previous substrate moisture strongly influences the hydrological performance of GR. As the inclusion of RCW in the substrate composition promotes a more effective drainage of the substrate during dry conditions, it is considered that RCW have a positive impact on GR’s hydrological performance.”
Point 6: Few places references are missing.
Thanks for your comment. We made a thorough bibliographic review resulting in the inclusion of more than 50 new references in the paper.
Once more, thanks for providing us with your valuable comments and suggestions that have indeed improved the quality of the manuscript.

Reviewer 2 Report
In this paper the authors simulate rainfall events and test the effect of substrate and plants for experimental scale green roofs in the Mediterranean climate. Currently, the majority of green roof literature comes from the Mediterranean, so this paper draws significant comparisons to other papers published under a similar climate type, however I do not think the appropriate comparisons were made in this article. After reading through the paper on different occasions, it is clear that some work needs to be conducted to get this article to a publishable standard. Currently, the manuscript reads as a well drafted paper, converted from a students experiment. The presentation of ideas should be streamlined to target the important aspects of the project to be published. There is also no discussion section presented, nor any discussion of the results section (no references). Please see below:
Line 38-51: The authors should consider writing the 3rd and 4th paragraph about BMPs instead of SUDS as their paper comes from the European Union, opposed to the United Kingdom.
Line 52-54: The justification seems a little weak as the majority of Green roof literature is based in the Mediterranean climate. I would consider taking this out of the introduction all together and perhaps including some more information on the climate type you’re assessing the experimental plots in to provide background of the research.
Line 61: Dividing the introduction into Chapters seems strange and is disjointed. The paper would benefit substantially from cutting this information down and creating one cohesive introduction section (as done by other paper published in Sustainability).
Lines 78-105: The inclusion of equations into the introduction detracts from the important message of the introduction. I think the introduction should be rewritten to be a single section that describes 1) the Mediterranean climate, 2) green roofs and their benefits, 3) hydrological performance of GRs in different climates, 4) hydrological performance of GRs in the Mediterranean, and then introduce your study.
The inclusion of tables in the introduction is good, however information is missing from the PDF review copy I received, therefore it is hard to assess the validity of these tables (i.e. I cannot see the last column “Number of…”. The inclusion of the tables is fine, but it feels as though these serve to justify the study, opposed to informing the audience of the purpose of the study.
Lines 131-132: Why is it reasonable to assume that GR can have a better hydrological performance in hot and dry summers compared to cold and humid winters? Depending on the substrate characteristics being too dry can make the substrate hydrophobic, which would reduce the performance. For the stated reasons here, please provide some reference to show evidence of this thought process.
The introduction could benefit from more of a literature review and compiling the information into a cohesive body of work. Several other research papers on the same topic provide much more background and are more thorough than the introduction provided here. 26 references, with two data tables in the introduction is quite light in respect to referencing. There is little to no information provided on green roof specifications and variation, let alone previous GR literature on topics outside of hydrology. For example, Lines 62-67 have no references, and then 3 references for lines 67-68 where you state the substrate depth. Please provide more background information on GRs in general. Also, the use of “State of The Art” is misplaced in this manuscript as you do not cover the state of the art, but simply a perspective on the current Mediterranean literature. This is very different to an actual “State of The Art review style.
Materials and Methods section is quite confusing, with much of the information written is a way that sounds like results. Please attempt to rewrite this section with only the important information that pertains to the set up and operation of the experiment. For example, the Line 169-171: does not need a description of the difference in nitric nitrogen concentration differences between substrate types. Additionally, the acronyms for the different treatment isn’t adequately explained, where as the reader I don’t know which treatment is which by the time I’m reading this section (Lines 160+).
Line 181: Figure 3: “main research steps…” I think the authors can remove this figure. The audience is aware of the research steps by reading the important parts of the methodology. The construction of the test beds etc is not an essential part of the methodology unless there is some temporal affect that needs to be explored.
Lines 185: if vegetation cover of the GR experimental test beds was collected, do you have LAI and change in LAI for each planted treatment?
Lines 218: the inclusion of the vegetation cover figure in the methodology is misplaced. This should be in the results and discussed appropriately. Does change in vegetation influence the results of the study? Please convert “vegetation cover” to LAI to standardise these results in respect to the rest of the international literature.
Results section 4.1: Vegetation cover. I do not believe this section is relevant to the paper as the assessment of vegetation cover was not the objective of your hydrological study. Please simply address the conversion of cover to LAI, determine if change in LAI was significant to the results recorded, state this and move onto the hydrological performance.
Figures 10-13 should be collapsed into a single figure that contains all four images. The current format is wasting space and difficult to observe.
Currently there is no discussion section of this article. The article in its current form is a good draft of a scientific article. Please include a discussion section, or rewrite the results to be “Results and Discussion” and include the appropriate referencing and discussion points.
The absence of a discussion section significantly limits the impact and suitability of this article for publication. 31 references for a hydrological paper is not suitable, and is significantly under assessed. The reviewer strongly recommends this article be rewritten with an entirely new perspective.
Author Response
Response to Reviewer 2 Comments
The authors wish to express gratitude to the reviewers for their time and effort in reviewing our manuscript.
As suggested by Reviewer 2, the paper was completely restructured and rewritten with a new perspective. Many changes were implemented, some to correct minor typographical errors, but most regarding the restructure of the manuscript. Chapter 1 (Introduction) was improved to include a general introduction and more background information on GR; to refer to other research papers and provide more information on GR specifications, variations, and benefits; and to introduce the important aspects of our study. A second chapter was created (2. Hydrological performance of GRs in the Mediterranean) referring to the Mediterranean climate and summarizing the existing studies on the hydrological performance of GRs in the Mediterranean.
Chapter 3 (Materials and Methods) was reorganized in three subchapters referring to site description, test beds characteristics, data collection and analysis and data analysis. This chapter presents the study site and experimental set up, the substrate characterization and composition, the methods and instruments used in the study, including their main limitations, and the variables considered to evaluate the hydrological performance of the installed GR.
Chapter 4 was rewritten to present the “Results and Discussion”, including the appropriate referencing and discussion points focusing on the effect of vegetation, substrate composition, previous moisture content and rainfall characteristics on the hydrological performance of GR.
Finally, in Chapter 5 (Conclusions) a summary of the study and it’s relevance is presented, evidencing the specific outcome of the investigation.
It should be noted that a thorough bibliographic review was done resulting in the inclusion of more than 50 new references in the paper.
As required, we have submitted a document entitled “compare doc.docx” that compares the initial and the final versions of the paper, with track changes turned on. In addition, due to the wide range of changes in the document, and to simplify the Reviewers understanding of the implemented modifications, we decided to prepare a file (“_sustainability-New V10_red changes.docx”) that consists of a final version of the paper, where the most important additions are highlighted in red text.
General aspects: In this paper the authors simulate rainfall events and test the effect of substrate and plants for experimental scale green roofs in the Mediterranean climate. Currently, the majority of green roof literature comes from the Mediterranean, so this paper draws significant comparisons to other papers published under a similar climate type, however I do not think the appropriate comparisons were made in this article. After reading through the paper on different occasions, it is clear that some work needs to be conducted to get this article to a publishable standard. Currently, the manuscript reads as a well drafted paper, converted from a students experiment. The presentation of ideas should be streamlined to target the important aspects of the project to be published. There is also no discussion section presented, nor any discussion of the results section (no references).
Thank for your valuable considerations. As mentioned, we agree with your perspective and therefore the paper was completely restructured and rewritten, mainly focusing on highlighting the important aspects of the project to be published and improving the discussion of the results, including more references.
Point 1: Line 38-51: The authors should consider writing the 3rd and 4th paragraph about BMPs instead of SUDS as their paper comes from the European Union, opposed to the United Kingdom.
Thanks for your comment. That paragraph was rewritten and focus was given to the BMP, as indicated below:
“The benefits from BMPs implementation in urban areas are diverse and include mitigation of flood risks (due to attenuation, retardation and infiltration), climate regulation (mitigation of urban heat island effects and heatwaves), and reduction of pollution loads (associated with stormwater and overflow discharges), while promoting biodiversity and the incorporation of socio-cultural services, including recreational areas, and focusing on landscape and human well-being principles [5, 6, 17].”
Point 2: Line 52-54: The justification seems a little weak as the majority of Green roof literature is based in the Mediterranean climate. I would consider taking this out of the introduction all together and perhaps including some more information on the climate type you’re assessing the experimental plots in to provide background of the research.
Thank you for your comment. This paragraph was reviewed. In addition, a second chapter was created (2. Hydrological performance of GRs in the Mediterranean) referring to the Mediterranean climate and summarizing the existing studies on the hydrological performance of GRs in the Mediterranean.
Point 3: Line 61: Dividing the introduction into Chapters seems strange and is disjointed. The paper would benefit substantially from cutting this information down and creating one cohesive introduction section (as done by other paper published in Sustainability)
Thanks for your comment. The introduction has been cut down and restructured to include a general introduction and more background information on GR; to refer to other research papers and provide more information on GR specifications, variations, and benefits; and to introduce the important aspects of our study. A second chapter was created (2. Hydrological performance of GRs in the Mediterranean) referring to the Mediterranean climate and summarizing the existing studies on the hydrological performance of GRs in the Mediterranean.
Point 4: Lines 78-105: The inclusion of equations into the introduction detracts from the important message of the introduction. I think the introduction should be rewritten to be a single section that describes 1) the Mediterranean climate, 2) green roofs and their benefits, 3) hydrological performance of GRs in different climates, 4) hydrological performance of GRs in the Mediterranean, and then introduce your study.
Thanks again for your valuable comment. The required modifications were made. The introduction has been rewritten, equations have been removed and a second chapter was added, as described in the previous point. We think we addressed all the reviewer suggestions.
Point 5: The inclusion of tables in the introduction is good, however information is missing from the PDF review copy I received, therefore it is hard to assess the validity of these tables (i.e. I cannot see the last column “Number of…”. The inclusion of the tables is fine, but it feels as though these serve to justify the study, opposed to informing the audience of the purpose of the study.
Thanks for your comment. There was a problem with the pdf formatting. We improved Table 1 and tried to evidence, in the preceding paragraphs, it’s relevance to the present study.
Point 6: Lines 131-132: Why is it reasonable to assume that GR can have a better hydrological performance in hot and dry summers compared to cold and humid winters? Depending on the substrate characteristics being too dry can make the substrate hydrophobic, which would reduce the performance. For the stated reasons here, please provide some reference to show evidence of this thought process.
Thanks for your comment. The following paragraph was added to the paper to support the idea, providing some references:
“In Mediterranean climatic conditions, rainfall is concentrated in the coldest season, often with significant precipitation intensity in short periods of time [59, 60, 50]. It is reasonable to assume that GR can have a better hydrological performance in hot and dry summers compared to cold and humid winters, since this performance is directly related to the previous degree of humidity of the substrate. Thus, in the rainy seasons (winter in the Mediterranean), the substrates will be frequently humid and close to saturation, and therefore will be able to retain less rainwater, resulting in a lower hydrological performance [78].”
Point 7: The introduction could benefit from more of a literature review and compiling the information into a cohesive body of work. Several other research papers on the same topic provide much more background and are more thorough than the introduction provided here. 26 references, with two data tables in the introduction is quite light in respect to referencing. There is little to no information provided on green roof specifications and variation, let alone previous GR literature on topics outside of hydrology. For example, Lines 62-67 have no references, and then 3 references for lines 67-68 where you state the substrate depth. Please provide more background information on GRs in general. Also, the use of “State of The Art” is misplaced in this manuscript as you do not cover the state of the art, but simply a perspective on the current Mediterranean literature. This is very different to an actual “State of The Art review style.
Thank you for your suggestions. As mentioned, the introduction has been rewritten and restructured, adding more information on GR and references to support it. The "State of Art" chapter was deleted; however, a second chapter ("Hydrological performance of GRs in the Mediterranean") was created to specifically refer to the hydrological performance of GR under Mediterranean climate.
Point 8: Materials and Methods section is quite confusing, with much of the information written is a way that sounds like results. Please attempt to rewrite this section with only the important information that pertains to the set up and operation of the experiment. For example, the Line 169-171: does not need a description of the difference in nitric nitrogen concentration differences between substrate types. Additionally, the acronyms for the different treatment isn’t adequately explained, where as the reader I don’t know which treatment is which by the time I’m reading this section (Lines 160+).
Thanks for your relevant comment. The required modifications has been implemented, so the “materials and methods” section was rewritten and reorganized. We believe that presently the procedures are clearer for the readers.
Point 9: Line 181: Figure 3: “main research steps…” I think the authors can remove this figure. The audience is aware of the research steps by reading the important parts of the methodology. The construction of the test beds etc is not an essential part of the methodology unless there is some temporal affect that needs to be explored.
Thanks for your comment. The figure has been removed from Materials and Methods.
Point 10: Lines 185: if vegetation cover of the GR experimental test beds was collected, do you have LAI and change in LAI for each planted treatment?
Thanks for your comment. In fact, the study of vegetation development was based on the percentage of vegetation cover (area occupied by plants/total area of the test bed), as referred by other authors, such as Liberalesso et al. (2021), Noya et al. (2021), Rocha et al. (2021) and Lonnqvist et al. (2021).
Point 11 and 12: Lines 218: the inclusion of the vegetation cover figure in the methodology is misplaced. This should be in the results and discussed appropriately. Does change in vegetation influence the results of the study? Please convert “vegetation cover” to LAI to standardise these results in respect to the rest of the international literature.
Results section 4.1: Vegetation cover. I do not believe this section is relevant to the paper as the assessment of vegetation cover was not the objective of your hydrological study. Please simply address the conversion of cover to LAI, determine if change in LAI was significant to the results recorded, state this and move onto the hydrological performance.
Thanks for your remarks. In fact, that figure was misplaced and was changed to the results and discussion section. As referred in the previous answer, the evaluation of vegetation development was based on the percentage of vegetation cover, following the approach of authors like: Liberalesso et al. (2021), Noya et al. (2021), Rocha et al. (2021) and Lonnqvist et al. (2021). We considered important to evaluate this parameter to establish if it was negatively influenced by the presence of RCW in the technical substrates. One of the study outcomes is the fact that the inclusion of RCW in the substrate composition did not hinder the development of vegetation or the hydrological performance of GR.
Point 13: Figures 10-13 should be collapsed into a single figure that contains all four images. The current format is wasting space and difficult to observe.
Thanks for your suggestion. We deleted those images and presented just one (Figure 8) to illustrate the effect of GR on peak flow attenuation.
Point 14: Currently there is no discussion section of this article. The article in its current form is a good draft of a scientific article. Please include a discussion section, or rewrite the results to be “Results and Discussion” and include the appropriate referencing and discussion points.
Thanks for your important comment. The topic was changed to Results and Discussions and further developed, as previously explained.
Point 15: The absence of a discussion section significantly limits the impact and suitability of this article for publication. 31 references for a hydrological paper is not suitable, and is significantly under assessed. The reviewer strongly recommends this article be rewritten with an entirely new perspective.
Thanks again for your comment. The suggestions were considered and the paper was completely rewritten, reorganized and improved, evidencing the relevance of the conclusions. Many references were added to support the discussion.
Once more, thank you for providing us with your valuable comments and suggestions that have indeed improved the quality of the manuscript.

Round 2
Reviewer 1 Report
The manuscript incorporated the suggestions, hence can be accepted.
Reviewer 2 Report
Significant improvement over the previous manuscript submitted. Thank you for taking the time to work on the document as what was required.